# Advances in Sintering Techniques for Calcium Phosphates Ceramics

**DOI:** 10.3390/ma14206133

**Published:** 2021-10-15

**Authors:** Abhishek Indurkar, Rajan Choudhary, Kristaps Rubenis, Janis Locs

**Affiliations:** 1Rudolfs Cimdins Riga Biomaterials Innovations and Development Centre of RTU, Institute of General Chemical Engineering, Faculty of Materials Science and Applied Chemistry, Riga Technical University, Pulka Street 3, LV-1007 Riga, Latvia; Abhishek-Rajesh.Indurkar@rtu.lv (A.I.); Rajan.Choudhary@rtu.lv (R.C.); kristaps.rubenis@rtu.lv (K.R.); 2Baltic Biomaterials Centre of Excellence, Headquarters at Riga Technical University, Kalku Street 1, LV-1658 Riga, Latvia

**Keywords:** calcium phosphates, sintering, bioceramics, bone tissue engineering

## Abstract

Calcium phosphate (CaP) biomaterials are extensively used to reconstruct bone defects. They resemble a chemical similarity to the inorganic mineral present in bones. Thus, they are termed as the key players in bone regeneration. Sintering is a heat treatment process applied to CaP powder compact or fabricated porous material to impart strength and integrity. Conventional sintering is the simplest sintering technique, but the processing of CaPs at a high temperature for a long time usually leads to the formation of secondary phases due to their thermal instability. Furthermore, it results in excessive grain growth that obstructs the densification process, limiting the application of CaP’s ceramics in bone regeneration. This review focuses on advanced sintering techniques used for the densification of CaPs. These techniques utilize the synergy of temperature with one or more parameters such as external pressure, electromagnetic radiation, electric current, or the incorporation of transient liquid that boosts the mass transfer while lowering the sintering temperature and time.

## 1. Introduction

Sintering is a thermal process in which loosely bound particles are converted into a consistent solid mass under the influence of heat and/or pressure without melting the particles. During the process, the atoms in the material migrate towards the boundaries of particles either by diffusion or assimilation. This results in the fusion of particles which leads to the formation of one solid piece. Sintering is a traditional technique known since human civilization and which was extensively used to treat (fire) pottery and earthenware [1]. Theoretical and experimental evidence of sintering came in the early 1900s. It is defined as a “processing technique used to produce density-controlled materials by utilizing thermal energy”. It is the most crucial step in ceramic manufacturing, wherein a green body is fired at a high temperature, below the melting point of the material. A green body is referred to as a self-supporting compact procured from powders that are compacted uniaxially or isostatically. During sintering, the raw material undergoes several physical and chemical transformations [2]. The sole aim of sintering is to enhance the product’s mechanical properties by particle bonding, densification, or recrystallization. For the accomplishment of desired mechanical properties’ materials, sintering is inevitable. Therefore, it is considered amongst the four essential components of material science and engineering [3].

CaP bioceramics belong to a resorbable and bioactive class of biomaterials and are widely used in bone regeneration and tissue engineering applications [4]. The specific formulation of CaP is illustrated in Table 1.

CaP has a chemical similarity to bones and teeth; therefore, it possesses inherent properties that support osteoblast adhesion and proliferation, and new bone formation [15]. As a result, they are employed in damaged or diseased mineralized tissue to reduce pain and restore function. The biomedical application includes the healing of bone defects, bone augmentation, fracture treatment, joint replacement, craniomaxillofacial reconstruction, orthopedics, spinal injury, ophthalmology, percutaneous devices, dental fillings, periodontal treatment, and dental implants [16]. The aim of tissue engineering is the de novo formation of tissues and organs. Significant efforts are taken in the development of 3D supports for cells that can stimulate the formation of an extracellular matrix [17]. The progress of calcium phosphate biomaterials in bone regeneration is elaborated elsewhere [18].

Although CaP bioceramics have excellent biocompatibility, low mechanical strength is a major hurdle for load-bearing applications. Therefore, for such applications, CaP must be consolidated at elevated temperatures. However, due to narrow temperature ranges of respective calcium phosphate consolidation at high temperatures, this leads to thermal instabilities which give rise to different phases. For example, tricalcium phosphate (TCP) is a well-known bioresorbable ceramic [19] that exists in four polymorphs, α, α’, β, and γ, that are stable at different temperatures. For instance, the α’-phase is stable at temperatures above 1470 °C whereas the α-phase is stable at temperatures ranging from 1125 to 1420 °C [20]. On the contrary, the β-phase is stable at lower temperatures (less than 1125 °C). The conversion of β→α→α’-TCP occurs when heat treatment is beyond the respective stability temperature. On the other hand, under the application of high pressure (4 GPa) and a high temperature (950 °C), β-TCP is converted to γ-TCP [21]. Among the polymorphs of TCP, α’ and γ phases are difficult to synthesize because of their high temperature and pressure requirements [22]. Due to the very narrow temperature range of the polymorphs, it is crucial to control the sintering temperature to obtain a pure phase of TCP.

Moreover, the amorphous TCP (also known as amorphous calcium phosphate (ACP)) converts rapidly to TCP at elevated temperatures (above 600 °C) [23]. Another form of CaP is stoichiometric hydroxyapatite (HA) which is stable at a temperature range from 1300 to 1550 °C. Exposure to higher temperatures leads to its conversion to α’-TCP and tetra-calcium phosphate [24]. In addition, a high sintering temperature and long holding time that is characteristic of conventional sintering lead to the coarsening of grains and surface contamination which deteriorates the mechanical properties of CaPs [25]. To overcome these drawbacks, advanced sintering techniques are developed that utilize either external pressure, electromagnetic radiations, electric current, or the incorporation of transient liquids that boost up the mass transfer while lowering the time and temperature. This review is focused on such advanced sintering techniques that have potential in the development of single-phase CaPs under lower temperatures than conventional sintering.

## 2. Sintering Process

The sintering process has three stages: initial, intermediate, and final. The powder characteristics are governing the parameters of the compact. The uniform arrangement of powders without flaws or defects gives rise to an ideal compact [26]. In the sintering process, the expression of changes in the connectivity of the particle within the ideal compact is shown in Table 2 [27]. Sintering is an irreversible process that takes place due to a reduction in Gibbs’s free energy. The three driving forces responsible for this reduction are: (i) surface curvature, (ii) external pressure, and (iii) chemical reaction [3]. Surface curvature is always the primary driving force of sintering. It is always present with or without the influence of pressure or a chemical reaction.

The addition of external pressure induces work conducted on the compact, which increases the driving force that aids the sintering process. The densification of a sample by a chemical reaction is the highest driving force compared to external pressure [28]. The detailed sintering behavior of these ceramics is illustrated elsewhere [29].

Densification and coarsening are two competitive mechanisms. Densification leads to shrinkage due to the grain boundary migration of the sintered object, while coarsening increases the particle size by interparticle mass transport, as shown in Figure 1. The reduction in surface or interface energy is the fundamental driving force of the sintering process indicated in Equation (1). γA stands for the total interfacial energy of the powder compact; γ is the specific surface area; and A is total surface area. In the equation, one Δγ is a change in interfacial energy, which leads to densification, while ΔA states a change in the interfacial area due to grain coarsening [30].
(1)Δ(γA)=(Δγ)A+γ(ΔA)

## 3. Conventional Sintering

Conventional sintering is the simplest technique by which the green body is heated up to a defined temperature without applying any external pressure, as represented in Figure 2A. It is also known as the “pressure-less sintering” technique [31]. Considering the thermal instabilities of CaPs, conventional sintering possesses a major drawback in fabricating pure phases of CaPs. In addition to this, treatment at high temperatures for longer times makes the process expensive and energy intensive. Grain growth occurs simultaneously with the densification process. This takes place actively during the final stage of sintering and is promoted at elevated temperatures. In conventional sintering, extensive grain growth at higher temperatures compromises the mechanical properties and densification. Therefore, it has become the major hurdle in the development of CaP materials for bone tissue engineering applications. Significant efforts in the development of advanced sintering techniques to achieve phase purity and dense materials are elaborated in the following sections.

### 3.1. Hot Pressing

Hot pressing (HP) is a sintering technique performed under the simultaneous application of heat and pressure [32]. As shown in Figure 2B, HP is committed to rigid graphite die by applying uniaxial pressure. The process is operated under a vacuum or protective atmosphere to avoid damage to the die. The uniaxial pressure delivers an extra driving force that elevates the sintering process by (a) increasing particle rearrangement in the initial stage, (b) accelerating the grain boundary sliding and diffusional creep in the intermediate and final stage, and (c) generating a surplus chemical potential and vacancy concentration gradient that promote mass transfer in the final stage [33]. Moreover, the friction resistance between the particles is reduced under heating and applied pressure to achieve dense packing. An external hydraulic system generates uniaxial pressure and radial pressure against the walls of the die [34]. The radial stress generated by applying pressure is given by the formula (v/1−v), in which v is the effective Poisson’s ratio. In the final stage of sintering, for nearly dense ceramics, v is equal to ½, where the radial stress comes close to the applied pressure. These radial stresses result in the acceleration of an interparticle shear leading to the particle rearrangement and collapse of the large pores. Therefore, acceleration in the initial stage of sintering is due to the creation of more particle contact. Grain boundary sliding can be accomplished by viscous flow in liquid or creep flow in solid-state sintering assisted by hot pressing. In dense ceramics’ gain, boundary sliding contributes to diffusional creep, assisting the pressure-aided sintering in the intermediate and final stages. Compared to conventional sintering, the consolidation of hydroxyapatite (HAp) by HP reveals a reduction in the temperature and holding time to 1100 °C and 30 min, respectively, under the pressure of 30 MPa [35]. Moreover, the addition of a sintering aid further reduces the requirement of a high temperature. Halouani et al. studied the effect of Na_3_PO_4_ on the densification of HAp by HP [36]. Results indicated that treatment at a temperature of 1000 °C within 30 min at a 20 MPa pressure results in a ceramic with a relative density of 97.5%. Compared to HP, in the absence of a sintering aid, the temperature requirement was increased to 1100 °C. Moreover, a drastic reduction in temperature was observed by combining hydrothermal synthesis with HP. HAp synthesized by hot hydrothermal pressing (HHP) required a temperature and pressure of 150 °C and 40 MPa, respectively, with a holding time of 6 h [37]. Moreover, a reduction in holding time to 3 h requires an increase in the pressure and temperature conditions up to 60 MPa and 200 °C, respectively [38].

### 3.2. Hot Isostatic Pressing

The hot isostatic pressing (HIP) technique utilizes the simultaneous application of isostatic pressure and heat for consolidation ceramics, as shown in Figure 2C. The primary factor that distinguishes HIP from other sintering techniques is the gas used for applying pressure in three dimensions to the material it surrounds [39]. For example, Argon gas heated to 1000 °C with 98 MPa of applied pressure results in higher heat transfer coefficients than conventional sintering. In HIP treatment, pressure, temperature, and time are the essential parameters that decide the density and mechanical properties of the product. The critical aspect of HIP is the isolation of the interface from applying pressure [40]. For sealing the contact area, the green body is encapsulated with thermally stable material. In HIP, heat is transferred through convection, and densification of the material occurs through particle rearrangement and plastic flow at the particle contact. This technique can work with up to 3000 °C and 200 MPa as the maximum temperature and pressure conditions [40]. Compared to uniaxial pressure in HP, isostatic pressure in HIP has flexibility in sintering samples having different shapes. One of the examples for the application of HIP is in the development of orthopedic implants, where fretting is observed due to potential degradation failure of the material [41]. Due to significant differences in the elastic modulus of two materials in contact, the shearing microenvironment occurs between the interface of the bone and implant. Fretting induces small groves and microgrooves in which the body fluid gets entrapped, leading to the corrosion of an implant. The oscillatory microenvironment at contact results in fatigue crack and wear, which cause early failure of the joint prosthesis [42]. Titanium (Ti) alloys are used as implants, but their biocompatibility and bioactivity are less than HAp. However, HAp is mechanically weak and cannot be used alone in load-bearing activities. Therefore, to overcome the drawback, HAp is coated on the implant surface [43]. HIP is extensively used for this process as it reduces the porosity and enhances ceramics’ physical and mechanical properties while maintaining the shape of the implant. In 1994, Hero developed HP coating on titanium alloys. Consolidation of the coating by HIP at a temperature of 700–850 °C and pressure of 1000 bars for 35 min was conducted [44]. Furthermore, Fu and co-workers successfully developed HP-coated titanium implants. Results indicate that HIP treatment leads to the densification of ceramics while decreasing the fretting wear [42].

### 3.3. Spark Plasma Sintering

Spark Plasma sintering (SPS) is also known as pulsed electric current sintering (PECS) or the Field-assisted sintering technique (FAST). It utilizes uniaxial pressure and pulsed direct electric current (DC) under low atmospheric pressure or inert gases to densify ceramics [45]. In general, SPS consists of a mechanical loading system, an electric current generator, a water-cooling system, and a vacuum apparatus which allow rapid heating and cooling rates. The rapid heating rates favor particles’ rearrangement, thus reducing the grain coarsening [46]. As shown in Figure 3A, SPS is a modification of the HP technique in which the application of the DC leads to Joule heating, achieving high heating rates of 1000 °C/min. The SPS system synchronizes both uniaxial pressure and the electric current to consolidate the sample [47]. Moreover, the consolidation of materials is accelerated by the application of pressure [48]. SPS takes place in four stages. Initially, the powder compact is between the graphite die, and a vacuum is generated by removing the gas. Then, uniaxial pressure is applied to the die which contains the powder, followed by resistant heating of the die and finally cooling [49]. This results in the heating of the sample from outside and inside, leading to self-heating. This rapid increase in the temperature is governed by self-heating and does not rely on the external heating source [50]. This enables the processing of compacts without degrading their inherent properties. Grain coarsening can be prohibited by SPS more than conventional powder sintering [51]. SPS provides an advantage in processing materials with thermal instabilities such as calcium phosphates. Gu and co-workers utilized the advantage of SPS and successfully demonstrated the sintering of pure HAp. The consolidation was performed at a temperature and pressure of 950 °C and 30 MPa, respectively. Further, a pulsed electric discharge was applied at 25 V and 750 A for 30 s. The actual densification took place when DC was used. After holding the sample for 5 min at the desired temperature, the pressure was released, and the electric current was terminated; further cooling was performed at 100 °C/min. The XRD analysis showed no phase difference between pre-sintered and sintered samples, which states that the consolidation by SPS preserved the intrinsic nature of the material. The relative density of the sintered HAp was 99.6%, which resembled HAp ceramics that were conventionally sintered at 1100 °C for 1 h [25,52]. In another study, Kawagoe et al. utilized SPS for the fabrication of dense β-TCP. This sintering was performed in 10 min at a temperature ranging from 800 to 900 °C, under a uniaxial pressure of 60 MPa and heating rate of 25 °C/min. The XRD analysis of the sintered product did not show any secondary phase, which states that the original properties of the materials were preserved after sintering. β-TCP prepared at 800 °C, 900 °C, and 1000 °C had 70, 95, and 99% relative densities, respectively. Moreover, treatment at 1000°C led to an improved transparency of β-TCP [53]. Compared to HIP and the conventional sintering of β-TCP, a reduction in temperature, pressure, and time was observed when the sintering was conducted by SPS. [49]. Nanocrystalline apatite with a preserved hydrated layer is tough to sinter by the conventional technique. The generation of higher pressure (100 MPa) is required to limit the water elimination from the structure. Furthermore, the application of a higher pressure limits the temperature requirement to 150–200 °C. SPS has the potential to develop biomimetic apatite with a preserved hydrated layer [54].

### 3.4. Flash Spark Plasma Sintering

Flash Sintering (FS) is a technique in which an electric field is applied to produce a dense product by decreasing the sintering temperature. This technique differs from SPS as it involves a high electric field, a shorter sintering period, direct current flow through the specimen, and a lower furnace temperature. Three mechanisms are proposed for FS: (a) grain boundary diffusion and electrical conductivity are enriched due to Joule heating at grain boundaries; (b) there is an increase in Frenkel pair nucleation due to the applied field; and (c) there is a change in self-diffusion at the grain boundaries due to the non-linear field applied [55]. FS is the latest technique employed in the fabrication of dense materials. This technique falls under the domain of field-assisted sintering techniques (FAST), which utilize the voltage and current of FS, and the cell design is similar to SPS [56]. FS is an energy-efficient process that involves a short time of 1–60 s. It operates in two steps; initially, the specimen is heated by a conventional sintering process, and then the current is applied over the heated sample to achieve rapid sintering. Figure 3B shows the graphical representation of flash spark plasma sintering. This thermal profile enables heating rates ranging from 250 to 9726 °C/min, completing the sintering in a fraction of seconds [57]. The FS starts with a fine powder with a density ranging from 10 to 50 mW/mm^3^. This technique was first used by Raj and co-workers who reported that the electric field retards the grain growth, resulting in an accelerated rate of sintering [58]. The consolidation of HA by FS was studied by Hwang and Yun. The powder compact was prepared by applying uniaxial pressure, followed by drilling two holes of a diameter of 1 mm. A platinum wire was hooked in the two holes, and contact resistance was avoided by sealing the holes by platinum paste. Further, the furnace was heated to 900 °C at a heating rate of 10 °C/min, followed by an application of an electric field over the heated sample. The occurrence of flash was detected in the range of a 925–1200 °C temperature and 500–1200 V/cm electric field. The XRD analysis of the sintered samples indicated the presence of pure HAp. The relative density of the FS samples that were sintered for 10 s at 1051 °C was compared with the samples that were conventionally sintered at 1192 °C for 5 min. The relative density of FS samples was superior to conventionally sintered samples. Studies also revealed that sintering in a vacuum was more beneficial than in air. At elevated temperatures, the dehydration of HAp leads to the generation of hydroxyl or hydrogen type defects, leading to an increase in the electric current during FS [59]. In conventional sintering, a higher temperature initiates the transition of the β phase to the α phase of TCP. This conversion follows first-order kinetics, which is controlled by a high activation energy. Therefore, the reconversion of α→β-TCP is kinetically inhibited on cooling, giving rise to a metastable α-TCP [60]. In conventional sintering, the conversion of β→α and α→α’ occurs at 1125 °C and 1470 °C, respectively [61]. To overcome this drawback, Frasnelli and Sglavo utilized FS for the consolidation of pure β-TCP without a transition to the α-TCP phase. The consolidation process was carried out in a horizontal loading dilatometer in which the axes of the green body were aligned to the Al_2_O_3_ boat with a constant heating rate of 20 °C/min. At the contact of the base surface of the sample, Pt-Rh electrodes were placed and connected to the DC supply. Further, the silver paste was applied to enhance electrical conductivity. The current was limited to 100 mA for less than 5 min. For comparison, the samples were also consolidated by conventional sintering at the same heating rate up to 1500 °C. Results suggested that FS requires less processing time as well as a lower temperature to achieve considerable shrinkage and avoid the β→α transition of TCP. Dense β-TCP was obtained at a temperature of 1000 °C in a few minutes [62].

### 3.5. Ultrafast High-Temperature Sintering

Ultrafast high-temperature sintering (UHS) is performed under a uniform high sintering temperature (up to 3000 °C) under high heating (10^3^–10^4^ °C/min) and cooling rates (10^4^ °C/min). The ultrafast temperature and heating rates enable rapid sintering (~10 s). UHS was first reported in 2020 by Wang et al., for the synthesis of Tantalum-doped Li_6.5_La_3_Zr_1.5_Ta_0.5_O_12_. In this study, the green body was sandwiched between Joule-heating carbon strips, which were rapidly heated by conduction and radiation, forming a uniform heating environment shown in Figure 3C [63]. Compared to microwave and spark plasma sintering, UHS required a simple setup, allowing for the consolidation of complex geometries. Pure phase β or α-TCP are tough to synthesize due to their thermal instabilities [21]. Biesuz et al. reported the successful synthesis of pure phase TCP by sintering calcium deficient hydroxyapatite (CDHA). Results indicated that CDHA transforms to polymorphs of TCP depending on the application of the current. The sample treated with 20–25A converted to β-TCP, whereas α-TCP was formed when more than a 30A current was applied. Depending upon the current application, the temperature is influenced, leading to different polymorphs of TCP. These observations suggested that by the application of UHS, it is possible to fabricate a single phasic TCP [24].

### 3.6. Microwave Sintering

Electromagnetic radiation in the range of 300 MHz to 300 GHz is termed microwaves. They are mainly used for telecommunication and heating purposes [64]. Based on electrical and magnetic properties, materials are classified into transparent, opaque, and absorbing [65]. The most crucial parameter for any material to interact with microwaves is permittivity or permeability. Microwave sintering (MWS) is a relatively new approach to sintering which differs from conventional sintering by the nature of the heat transfer as shown in Figure 4A. In conventional sintering, the heat is transferred to materials from the surface to the core, while in MWS, the process occurs through several physical mechanisms such as resistive heating, bipolar rotation, and electromagnetic and dielectric heating. One or more mechanisms can be attributed to the response to the incoming radiation depending on the material [66]. When microwaves penetrate through the material, the electromagnetic waves stimulate motion in free and bound charges and dipoles. The natural equilibrium of the material resists the motion due to frictional, inertial, and elastic forces, which cause the dissipation of energies. This leads to an attenuation of the electric field associated with microwaves, and heating of the material occurs [67]. Therefore, MWS is categorized as a non-conventional technique of sintering. It is a fast, reliable, and user-friendly technique, with the significant advantage of higher heating rates, lower energy consumption, and lower cost. Hong Sung and associates studied the kinetic mechanism of MWS in ceramic materials and stated that the densification mechanism in MW is different from that of conventional sintering. Their model indicates that the MWS shrinkage rate is proportional to t^2/3^, whereas the traditional shrinkage of the sintering rate is proportional to t^2/5^, revealing that MWS is faster than conventional sintering [68]. Utilizing the advantages, Fang and co-workers studied the effect of MWS on the densification of hydroxyapatite (HAp). The experiment was performed in a 500 W microwave in which the specimen was sintered at temperatures ranging from 1200 to 1300 °C for different time points of 5, 10, and 20 min, respectively. The HAp pellets attained a density of 97% after sintering for 10 min. Compared to the convention sintering, MW consumed less energy and offered a dense structure, with a finer grain size and improved mechanical strength [69]. The consolidation of TCP by MWS was performed by Mirhadi. The green body was heated to the three different temperatures of 900 °C, 1000 °C, and 1100 °C, respectively, at a 20 °C/min heating rate. The XRD analysis of the sintered body revealed the presence of a pure β-TCP phase. The relative density of samples treated at 1100°C was found to be 98%, whereas the relative density of 81% was observed when samples were treated at the same temperature by conventional sintering for 2 h [70].

### 3.7. Laser Sintering

Laser sintering (LS), also known as selective laser sintering, was developed by Carl Deckard in 1980 [71]. It is an additive manufacturing technique that utilizes a high-power laser to fuse small particles of material layer-by-layer to form a component of a defined architecture. Industries widely accept this technique to fabricate complex parts in a single operation [72]. The LS system consists of a laser source, powder recharging system, building platform, and gas flow controller as shown in Figure 4B [73]. The process is governed by computer-aided manufacturing (CAM). Depending on the photon absorption characteristics of the materials, two types of lasers are used in LS, a continuous wave CO_2_ laser (10.6 μm) and a yttrium fiber laser (1064 nm) [74]. The LS process begins with converting 3D CAD data into slices of STL files fed to the machine. Depending on the architecture of the STL file, the computer controls the scanning track of the laser. Subsequently, the sample is heated up just below the melting point of the material. The high-energy laser beam is guided by the CAD/CAM design to fuse the powder in a layer-by-layer pattern. After treating each layer, the powder bed is lowered by one layer of thickness, and fresh powder is rolled over the platform [75]. Both the rapid heating of the powder and cooling of melts occur when the laser moves quickly over the powder. Therefore, LS can be termed as a “high power density short interaction time” process. The melting/solidification approach is the mechanism of densification, following first-order kinetics. The laser energy input and powder characteristics play a significant role in densification kinetics. A higher sintering rate is observed in finer particles, as they possess a higher surface area which provides more energy absorbed from the laser [76]. Ceramic compounds have very high melting points; therefore, the energy inputs are very high. A continuous-wave CO_2_ laser has a high-energy beam that is absorbed by most of the ceramics [77]. LS can be performed directly. In this case, the laser is used to consolidate powder locally, and in the indirect mode, a sintering agent is deployed to enhance the densification. Qin and associates performed experiments for the development of tetra tricalcium phosphate (TTCaP) scaffold by selective LS. The parameters were as follows: laser power was varied from 6 to 10 W; the laser beam diameter was 1 mm; the line scanning speed was kept to 60 mm/s; the layer thickness was 0.1 mm; and the laser line thickness was 2 mm. Results indicate that only a few particles melted and fused at a laser power of 6 W due to insufficient energy. By increasing the laser power to 7 W and 8 W, more particles were fused, reducing the micro gaps. At a further increase in laser power to 9 W and 10 W, a compact structure was obtained without micro gaps. The XRD analysis of the sintered sample at a 9 W laser power shows a pure phase of TTCP without any impurities [78]. Bulina et al. studied the laser sintering of HAp, and a CO_2_ laser was used at 4 W with a spot size of 0.2 mm and scanning speed of 640 mm/s. The fast-heating rates lead to the melting of HAp in the air without decomposition, which results in dense HAp with a preferential orientation in the c-axis. The XRD analysis showed a minor presence of β-TCP, which occurs because of very high heating [79]. Qingxi and co-workers attempted to sinter β-TCP by LS in which additives such as epoxy and nylon for the adhesion of β-TCP were used. The composite was formed by the physical mixing of additives with β-TCP. LS was performed at room temperature with a laser power of 11 W with a 0.15 focused diameter and scanning speed of 2400 mm/s, and a layer thickness of 0.1 mm. After that, the specimen was treated at 600 °C and further sintered at 1100 °C for 3 h to rid the specimen of binders. The XRD analysis of the treated samples matches with the standard β-TCP phase and shows the presence of HAp and α-TCP [80].

### 3.8. Cold Sintering

Cold sintering (CS) is a new technique that utilizes low temperatures (up to 300 °C) to consolidate ceramic powders. All the other densification techniques discussed above require higher temperatures for densification of the material. This not only reduces the energy needed but also allows for the consolidation of heat-sensitive materials and composites, which cannot be sintered using other techniques [81]. CS has opened new horizons for the sintering of hybrid materials such as organic/inorganic components. In 2016, Guo and co-workers demonstrated the CS process for the first time; since then, it has attracted many researchers [82]. CS is a pressure-assisted method carried out in the presence of a transient liquid. The process is performed either at room temperature or an elevated temperature up to 300 °C, depending on the boiling point of the liquid. Typically, in CS, uniaxial pressure is applied to powder/liquid mixtures [83]. Densification of the powder is driven by a dissolution and reprecipitation process coupled with Oswald ripening and a recrystallization phenomenon [84]. CS takes place in two stages, as shown in Figure 4C. In the first stage, the application of pressure to the liquid medium generates isostatic pressure (irrespective of the pressure applied uniaxially or isostatically), leading to four main phenomena: (a) initially, the material is partially dissolved in a liquid medium; (b) Ostwald ripening increases the average particle size; (c) recrystallization may occur (the formation of new crystals or phases in the liquid); and (d) drying is initiated. Based on drying, the uniaxial CS and isostatic CS are different. Dying in uniaxial CS takes place through the gap clearance; conversely, samples must be withdrawn from the press for drying in the isostatic CS. After that, the second stage starts, where solid particles make contact and give rise to a rigid framework. Three phenomena take place: (a) particle rearrangement, (b) solvent evaporation, and (c) precipitation. In addition to this, if pressure is maintained in stage II, the fourth phenomenon occurs; sintering leads to mass transfer and particle shape modification [85]. A recent development in CS occurred, resulting in the densification of materials which was not possible by other techniques. Amorphous calcium phosphate (ACP) is challenging to sinter because of its unique hydrated structure and conversion to TCP or HAp at higher temperatures. Owing to the advantage of sintering samples at low temperatures, Rubenis et al. employed CS to densify ACP. The powder (ACP) was moistened with 0.1 mL of deionized water before pressing to study the effect of the transient liquid on CS. Further, a uniaxial pressure of 500 MPa was applied to the powder and sintered at room temperature, 100 °C, 120 °C, and 150 °C, respectively, with a holding time of 30 min. The XRD analysis revealed that the amorphous state was retained at room temperature, 100 °C, 120 °C when ACP was in a dry state. In contrast, in the moistened form, only treatment at room temperature retained the amorphous state. A further increase in temperature leads to the conversion of the amorphous state to crystalline. A relative density of ~76%, ~79%, ~76%, and ~85% was obtained when the sintering was performed at room temperature, 100 °C, 120 °C, and 150 °C, respectively [86]. Hassan, Akmal, and Ryu investigated the effect of vacuum-oven-dried HAp (110 °C) and calcinated HAp powder (1000 °C) on CS. The experiment was performed at a pressure of 500 MPa at 200 °C. Post sintering, the relative density of 98.8% was achieved, and the XRD analysis revealed a pure phase of apatite. The chemical-mechanical phenomenon drives low-temperature sintering. Therefore, the stoichiometry and the mechanical properties were not influenced by CS [87]. Owing to the advantage of the low-temperature processing of CS, it can effectively be used in the fabrication of organic/inorganic composites for bone tissue engineering.

## 4. Discussion

CaPs are the integral component of tissue engineering. The sintering of CaPs is vital to achieve desired mechanical properties. However, CDHA transforms to β-TCP or a biphasic β-TCP/HAp mixture below 1000 °C, whereas β-TCP is converted to α-TCP around 1125 °C, and a further increase in temperature leads to the formation of α’-TCP (1470 °C). Similar effects are observed when ACP and HAp are treated at very high temperatures for a long time. Moreover, a transformation of β→α-TCP at a higher temperature and holding time leads to volume expansion, thus hampering the densification process. Innovative technologies utilized various parameters, shown in Table 3, to overcome the limitations of conventional sintering. These techniques include HP, HIP, CS, MW, SPS, LS, UHS, and FS. CS, SPS, and FS are not capable of sintering structures with complex architectures, whereas MW, LS, HIP allow the sintering of structures with complex architectures.

The instrumental setup of conventional sintering is relatively simple and allows for the sintering of a large number of samples simultaneously. However, the high-temperature requirement and long holding time make the operation very expensive and time-consuming. The major advantages of conventional sintering are scalability and the ability to process a large number of samples. The pressure-assisted sintering techniques reduce the sintering time by accelerating the rate of mass transfer. This approach differs in the stress state during density which influences particle sliding and the change in dimension. The rate of densification and shrinkage is governed by the application of pressure [88]. Moreover, grain growth is inhibited by decreasing diffusivity and thus grain boundary mobility. The applied pressure is independent of the particle size, whereas the intrinsic sintering pressure increases when the particle size is reduced [89]. The power creep law is an effective mechanism of densification in pressure-assisted sintering [90]. This confirms that with pressure, densification of the material requires a temperature and time lower than conventional sintering. The techniques falling under pressure-assisted sintering are HP, HIP, SPS, FS, and CS. This advancement enabled the fabrication of pure dense materials, for example, the fabrication of dense ACP by CS, pure β-TCP by SPS and FS, which is not possible by conventional sintering. In addition, this advancement led to this considerable reduction in sintering time from hours to minutes. A significant limitation of pressure-assisted sintering techniques is dealing with a sample of industrial-relevant sizes. As the sample size increases, the requirement for equipment size and electric consumption also increases, making the process expensive. Furthermore, in SPS and FS, increases in the sample diameter elevate the current requirement for a high heating rate beyond 100 °C/min. Moreover, the increased need for achieving a high temperature and high heating rates makes the process more expensive on an industrial scale [49].

Field-assisted sintering technologies (FAST) utilize an electric current with a synergy of temperature and pressure. The application of an electric current provides a uniform temperature gradient and additional mass transfer, reducing the time and temperature requirements. The techniques that fall under FAST are SPS, FS, and UHS. The sintering time by these techniques ranges from minutes to seconds. MW and LS are two classes of sintering that utilize electromagnetic radiation for heat transfers. This creates higher heating rates and a uniform heat distribution that accelerates the densification process. They also have the potential to consolidate materials with complex architectures. Due to the non-uniformity of the laser route, the heat formation is non-homogenous, and the limitation of the laser diameter does not allow for the fabrication of objects with exemplary architectures. Compared to conventional sintering, MW and LS have very sophisticated instrumentation, which makes the entire process expensive. Though there are few limitations to the advanced sintering techniques, they can potentially sinter pure CaP compounds that can be effectively incorporated in tissue engineering applications.

## 5. Summary and Concluding Remarks

Sintering is a thermal process in which loosely bound particles are converted into a consistent solid mass under the influence of heat and/or pressure without melting the particles. During this process, grain growth and densification occur simultaneously. Typically, this is observed in the last stage of the sintering process and is promoted by high temperatures. Densification of CaP ceramics by conventional sintering is an energy-intensive and time-consuming operation. Due to the thermal instability of CaP, treatment at a high temperature for a long time leads to the formation of secondary phases. Furthermore, it also results in excessive grain growth that obstructs the densification process. To overcome these drawbacks, substantial efforts have been dedicated to developing innovative sintering techniques. These techniques utilize temperature in synergy with one or more parameters such as external pressure, radiation, electric current, the incorporation of transient liquid, etc. This accelerates the mass transfer, and therefore a considerable reduction in temperature and time is observed. A comparative analysis of the time and temperature required by advanced sintering techniques and conventional sintering is shown in Figure 5. This indicates that advanced sintering techniques considerably reduce temperature and time requirements. These techniques perform exceptionally well on a small scale, but the translation to the industrial scale remains a challenge. Apart from these minor limitations, advanced sintering techniques have potential in the development of organic/inorganic composites and pure phases of CaP that was not possible by conventional sintering. This advancement has opened new doors in the development of pure materials for bone tissue regeneration.

## Figures and Tables

**Figure 1 materials-14-06133-f001:**
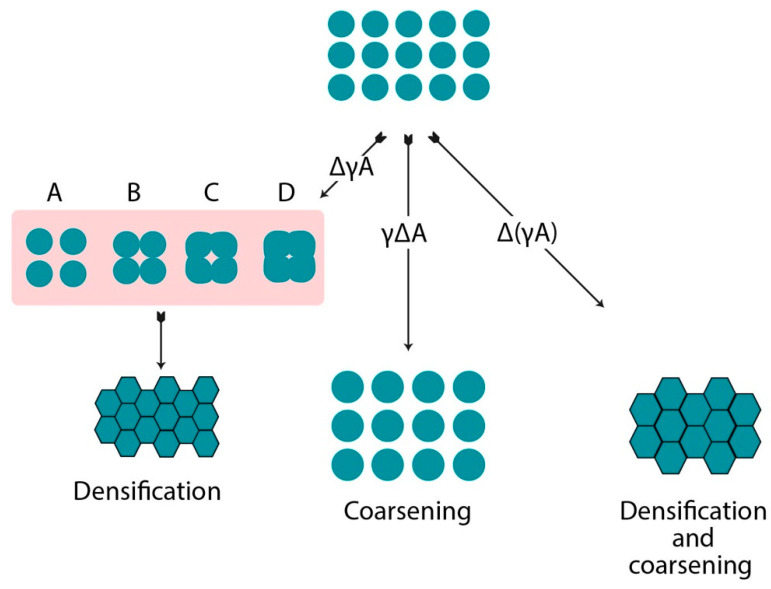
A basic phenomenon occurring during the sintering process.

**Figure 2 materials-14-06133-f002:**
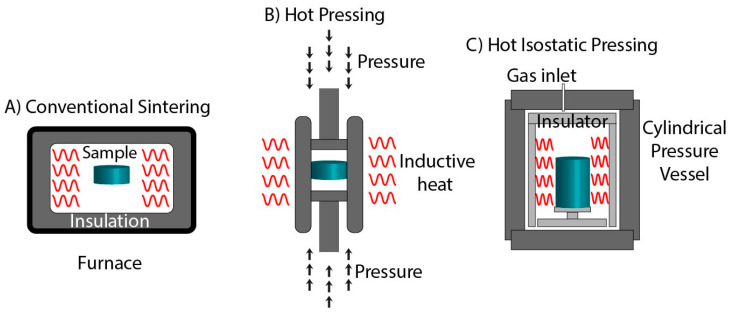
Schematic representation of (**A**) Conventional sintering, (**B**) Hot pressing, and (**C**) Hot Isostatic pressing.

**Figure 3 materials-14-06133-f003:**
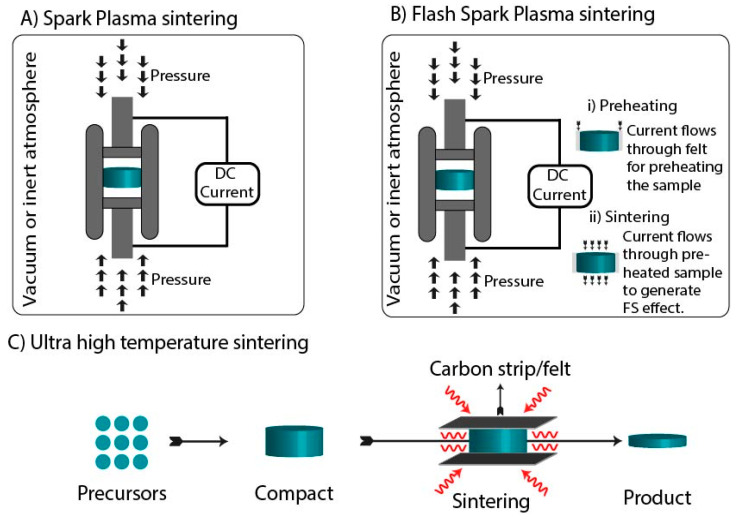
Schematic representation of (**A**) Spark Plasma sintering, (**B**) Flash spark plasma sintering, and (**C**) Ultrafast high-temperature sintering.

**Figure 4 materials-14-06133-f004:**
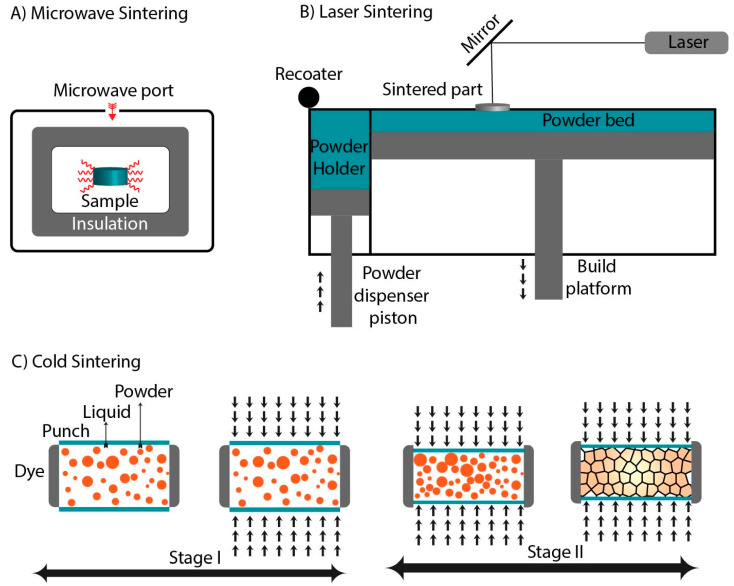
Schematic representation of (**A**) Microwave sintering, (**B**) Laser sintering, and (**C**) Cold sintering.

**Figure 5 materials-14-06133-f005:**
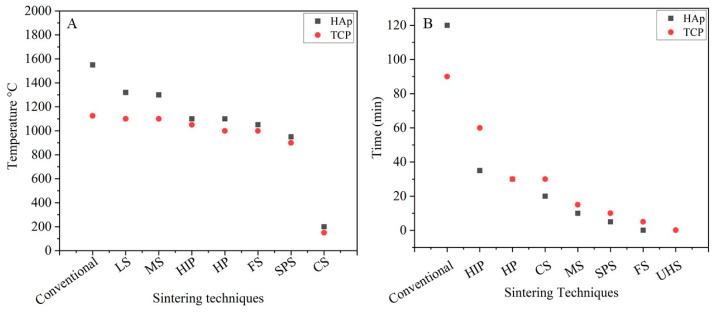
Comparison of sintering (**A**) temperature and (**B**) time required for conventional and advanced sintering techniques.

**Table 1 materials-14-06133-t001:** Physical properties, synthesis, and application of various types of calcium phosphates. Adapt from references [5,6,7,8,9,10,11,12,13,14].

Commercial Products	Implant Form	Biological Occurrence	Synthesis	Density (g/cm^3^)	−log K_sp_ at 25 °C	Melting Point (°C)	Ca/P Molar Ratio	Crystal Structure	Chemical Formula	Phase
Bone source (TTCaP and DCP)	Cement	-	Solid-state reaction at temperature 1450–1500 °C for 6–12 h.	3.05	38	Decomposes and transforms to HAp	2	Monoclinic	Ca_4_(PO_4_)_2_O	Tetra-calcium phosphate (TTCaP)
Ostim (HA) Graftys HBS (HA and TCP) Graftys quickset (HA and TCP)	Coating or deposited with polymer	Soft tissue calcification, enamel, dentin, bone, tooth, and urinary calculus	Precipitation performed at pH 9.5–12 at 90 °C	3.16	58.4	1670	1.67	Pseudohexagonal	Ca_10_(PO_4_)_6_(OH)_2_	Hydroxyapatite (HAp)
Fracture grout (α-TCP and Calcium carbonate CaCO_3_) Biopex (α-TCP, DCPD, HA, and TTCaP)	Block	Dental and urinary calculus, salivary stones, tissue calcification, and milk Magnesium substituted β-TCP is identified in soft tissue calcification and dental calculus	Thermal Decomposition of β-TCP above 1125 °C	2.86	25.5	1391	1.5	Monoclinic	Ca_3_(PO_4_)_2_	Tricalcium phosphate (a-TCP)
ChronOS Inkjet (β-TCP and DCPD)	Granules or block		a. Solid-state reaction b. Decomposition of calcium-deficient HAp above 750 °C c. Precipitation in an organic solvent	3.08	28.9	Decomposes and transforms α-TCP	1.5	Rhombohedral	Ca_3_(PO_4_)_2_	Tricalcium phosphate (β-TCP)
-	Granules	Dental and urinary calculus	Precipitation performed at pH 5.5–7.0	2.61	96.6	Decomposes and transforms to HAp	1.33	Triclinic	Ca_8_H_2_(PO_4_)_6_·5H_2_O	Octa-calcium phosphate (OCP)
Embarc (ACP and DCPD) α−ΒΣΜ (AΧΠ ανδ ΔΧΠΔ)	Coating or granules	Kidney stone and heart calcification in uremic patients and soft tissue calcification	Precipitation performed at pH 5.0–12.0	0.92–1.75	26–33	Transforms to a stable phase	1.2–2.2	Amorphous	Ca_3_(PO_4_)_2_	Amorphous calcium phosphate (ACP)
Eurobone (DCPD and TCP) Calcibon (DCPD, α-TCP, CaCO_3_ and HA)	Powder	Dental calculi, chondrocalcinosis, crystalluria, and carious lesions	Precipitation performed below 80 °C under pH 2.0–6.5	2.32	6.59	Decomposes and transforms to a stable phase	1	Monoclinic	CaHPO_4_·2H_2_O	Dicalcium phosphate dihydrate (DCPD)

**Table 2 materials-14-06133-t002:** Stages in the sintering process.

Stage	Process	Densification	Coarsening	Loss in Total Specific Surface Area (SSA)
Initial	Neck growth formation	Trivial	Negligible	50%
Intermediate	Elongation and rounding of pores	Substantial	Increase in grain and pore size	Complete loss of open porosity
Final	Closure of pores with final densification	Slow and minor change in density	Enormous pore and grain growth	Negligible

**Table 3 materials-14-06133-t003:** Parameters of various sintering techniques.

Technique	Temperature	Pressure	Current	Electromagnetic Radiations
Conventional sintering	✓	x	x	x
Hot Pressing	✓	✓	x	x
Hot Isostatic Pressing	✓	✓	x	x
Spark Plasma sintering	✓	✓	✓	x
Flash Spark Plasma sintering	✓	✓	✓	x
Ultrafast high-temperature sintering	✓	x	✓	x
Microwave sintering	✓	x	x	✓
Laser Sintering	✓	x	x	✓
Cold sintering	✓	✓	x	x

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
