# Peer review of "Advances in Sintering Techniques for Calcium Phosphates Ceramics"

_materials, 2021, doi:10.3390/ma14206133_

Round 1
Reviewer 1 Report
This paper contains logical systematization of the sintering techniques. I belive, that it will be interesting for a lot of readers in the field of ceramic materials.
Author Response
Reviewer 1 - Comments and Suggestions for Authors
This paper contains logical systematization of the sintering techniques. I believe that it will be interesting for a lot of readers in the field of ceramic materials.
Response – No commentsReviewer 2 Report
- Page 2, in the sentence: "Due to the very narrow temperature range of the polymorphs, it is crucial to control the sintering temperature to obtain a pure phase."
- “Pure phase” refersa to CaP or TCP? It would be interesting for the authors to make this clear in the text;
- I suggest that authors include the chemical formulas of substances in the text;
- Many ideas in the text are not clear and this is in many parts of the text because the ideas are too short. For example, in the Page 12: "The major advantage of conventional" - what does it refer to?
- In the topic “5. Future Directions” are presented a lot of information in a superficial way, which ends up not making clear the understanding of this part.
- Can the authors explain the function of the item “6.Summary”? I couldn't understand the function of this item in the text.
- Figure 5a-b relates to item “6. Summary”. However, the substances shown in these figures are HAP and TCP. Can the authors explain this fact?
Author Response
Reviewer 2 - Comments and Suggestions for Authors
- Page 2, in the sentence: "Due to the very narrow temperature range of the polymorphs, it is crucial to control the sintering temperature to obtain a pure phase." “Pure phase” refers to CaP or TCP? It would be interesting for the authors to make this clear in the text.
Response – The authors have corrected the sentence in the revised manuscript as “ Due to very narrow temperature range of polymorphs, it is crucial to control sintering temperature to obtain pure phase of TCP.”
- I suggest that authors include the chemical formulas of substances in the text.
Response – A detailed table is included in the introduction that covers the chemical formula, physical characteristics, preparation process, biological occurrence, and examples of commercial products. In addition to this the biomedical importance of CaP ceramics in bone regeneration is included.
- Many ideas in the text are not clear and this is in many parts of the text because the ideas are too short. For example, in the Page 12: "The major advantage of conventional" - what does it refer to?
Response – Author has rectified the mistake, the sentence was completed, “The major advantage of conventional sintering is scalability and ability to process large number of samples.”
- In the topic “5. Future Directions” are presented a lot of information in a superficial way, which ends up not making clear the understanding of this part.
Response – Author has clubbed the section of future direction and summary to one section summary and concluding remarks with detailed information.
- Can the authors explain the function of the item “6. Summary”? I couldn't understand the function of this item in the text.
Response – The information in the summary section is looking repetitive. Therefore, the section of future direction and summary was clubbed in to one summary and concluding remarks.
- Figure 5a-b relates to item “6. Summary”. However, the substances shown in these figures are HAP and TCP. Can the authors explain this fact?
Response – Majority of studies are performed on TCP and HAP; it is explored in all the advanced sintering techniques. For other calcium phosphates expect HAP and TCP the studies are not yet available. Therefore, in Fig 5 the comparison of time and temperature of conventional and advanced sintering techniques reported for TCP and HAP. The main aim of these graphs are to highlight the advancement in sintering techniques.
Reviewer 3 Report
This review presents “Advances in sintering techniques for calcium phosphates ce-ramics”. The results achieved in the introduced manuscript is important. This manuscript is recommended to be published after including and addressing the below listed comments with minor corrections.
- The authors should eliminate the current grammatical and punctuation mark errors and also confirm the correct scientific English.
- The authors should write the complete terms of all abbreviations (including the instruments) before the first use in the abstract and main manuscript.
- The authors should revise the introduction of the manuscript to include the importance and advantages of the proposed ceramics and preparation processes.
- The authors should cite important references related to the biomedicine.
- The below reference are suggested be cited on the revised manuscript:
Acta Biomaterialia Volume 9, Issue 4, April 2013, Pages 5855-5875
International Journal of Refractory Metals and Hard Materials 95, 105444 (2021)
Journal of the European Ceramic Society Volume 41, Issue 1, January 2021, Pages 912-919
- The summary section should discuss the perspectives, challenges, future applications and aspects of the similar ceramics.
Author Response
Reviewer 3 - Comments and Suggestions for Authors
This review presents “Advances in sintering techniques for calcium phosphates ceramics”. The results achieved in the introduced manuscript is important. This manuscript is recommended to be published after including and addressing the below listed comments with minor corrections.
- The authors should eliminate the current grammatical and punctuation mark errors and confirm the correct scientific English.
Response – The authors have revised the manuscript.
- The authors should write the complete terms of all abbreviations (including the instruments) before the first use in the abstract and main manuscript.
Response – Authors have included full list of abbreviation before the first use in the abstract and main manuscript.
- The authors should revise the introduction of the manuscript to include the importance and advantages of the proposed ceramics and preparation processes.
Response – A detailed table is included in the introduction that covers the chemical formula, physical characteristics, preparation process, biological occurrence, and examples of commercial products. In addition to this the biomedical importance of CaP ceramics in bone regeneration is included.
- The authors should cite important references related to the biomedicine.
Response – The references of biomedical application of CaP’s are included.
Samavedi, S.; Whittington, A.R.; Goldstein, A.S. Calcium phosphate ceramics in
bone tissue engineering: A review of properties and their influence on cell behavior.
Acta Biomater. 2013, 9, 8037–8045, doi:10.1016/J.ACTBIO.2013.06.014.
Graça, M.P.F.; Gavinho, S.R. Calcium Phosphate Cements in Tissue Engineering.
Contemp. Top. about Phosphorus Biol. Mater. 2020,doi:10.5772/INTECHOPEN.89131.
Canillas, M.; Pena, P.; De Aza, A.H.; Rodríguez, M.A. Calcium phosphates for
biomedical applications. Boletín la Soc. Española Cerámica y Vidr. 2017, 56, 91–112,
doi:10.1016/J.BSECV.2017.05.001.
Jeong, J.; Kim, J.H.; Shim, J.H.; Hwang, N.S.; Heo, C.Y. Bioactive calcium
phosphate materials and applications in bone regeneration. Biomater. Res. 2019 231
2019, 23, 1–11, doi:10.1186/S40824-018-0149-3.
- The below reference is suggested be cited on the revised manuscript:
- Acta Biomaterialia Volume 9, Issue 4, April 2013, Pages 5855-5875
Response – Author has included reference in the manuscript
- International Journal of Refractory Metals and Hard Materials 95, 105444 (2021)
Response - Author has included reference in the manuscript
- Journal of the European Ceramic Society Volume 41, Issue 1, January 2021, Pages 912-919
Response - Author has included reference in the manuscript
- The summary section should discuss the perspectives, challenges, future applications, and aspects of the similar ceramics.
Response – Author has modified these sections as per the reviewers comments.
Round 2
Reviewer 2 Report
After the authors' corrections, I consider that the article can be accepted for publication.
Reviewer 3 Report
The manuscript is revised properly. It can be accepted for publication after approve of the editor.